# LANGUAGE-ENHANCED REPRESENTATION LEARNING FOR SINGLE-CELL TRANSCRIPTOMICS

**Jiaqi Yang**[1], **Zhiyuan Liu**[2], **Yaorui Shi**[1], **Qingchuan Zhang**[1], **Changhao Nai**[3],
**Junfeng Fang**[2], **Yahui Long**[4], **Yang Zhang**[2], **Xiang Wang**[1]
[1]University of Science and Technology of China   [2]National University of Singapore
[3]Harbin Institute of Technology   [4]Central South University
{acharkq,yaoruishi,zhangqingchuan32,naichh12,xiangwang1223}@gmail.com
{yangjiaqi20,fjf}@mail.ustc.edu.cn, longyahui@csu.edu.cn, zhang@nus.edu.sg

## ABSTRACT

Single-cell RNA sequencing (scRNA-seq) offers detailed insights into cellular heterogeneity. Recent advancements leverage single-cell large language models (scLLMs) for effective representation learning. These models focus exclusively on transcriptomic data, neglecting complementary biological knowledge from textual descriptions. To overcome this limitation, we propose scMMGPT, a novel multimodal framework designed for language-enhanced representation learning in single-cell transcriptomics. Unlike existing methods, scMMGPT employs robust cell representation extraction, preserving quantitative gene expression data, and introduces an innovative two-stage pre-training strategy combining discriminative precision with generative flexibility. **Extensive experiments across nine datasets and five tasks show that scMMGPT significantly outperforms unimodal and multimodal baselines**. Notably, scMMGPT achieves relative improvements of **8.13%** for clustering on the COVID-19 dataset, **7.48%** for annotation on the Myeloid dataset, and **2.93%** for perturbation prediction on the Adamson dataset. Our code is available at https://github.com/segmenttree9/scMMGPT.

## 1 INTRODUCTION

Single-cell RNA sequencing (scRNA-seq) profiles gene expression at the level of individual cells, providing a fine-grained view of cellular heterogeneity (Saliba et al., 2014; Shalek et al., 2014; Silverman et al., 2020). The complexity and high dimensionality of scRNA-seq data necessitate powerful computational approaches that can leverage massive datasets efficiently and accurately (Angerer et al., 2017). Inspired by the success of large language models (LLMs) in natural language processing (Vaswani et al., 2017; OpenAI, 2023; Touvron et al., 2023a), specialized single-cell large language models (scLLMs) have emerged, leveraging self-supervised pre-training on extensive expression datasets to produce robust cell representations for downstream tasks such as cell annotation and clustering (Theodoris et al., 2023; Cui et al., 2024; Yang et al., 2022; Hao et al., 2024). However, existing scLLMs are mostly pre-trained solely on scRNA-seq data, inherently constraining the breadth of their cell representations.

In this work, we explore **language-enhanced single-cell representation learning**, aiming to integrate the fine-grained molecular signals from scRNA-seq with the high-level biological knowledge encoded in textual descriptions and metadata. As Figure 2 shows, textual descriptions encode contextual information–such as species, tissue origin,

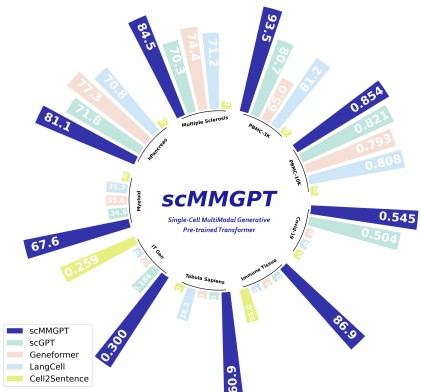

Figure 1: Performance of various scLLM on single cell tasks. We demonstrate results in cell type annotation, cell clustering, batch effect removal, cell description generation and pseudo-cell generation task. Detailed descriptions for metrics are provided in §E.4. Best viewed in color.

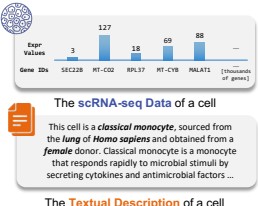

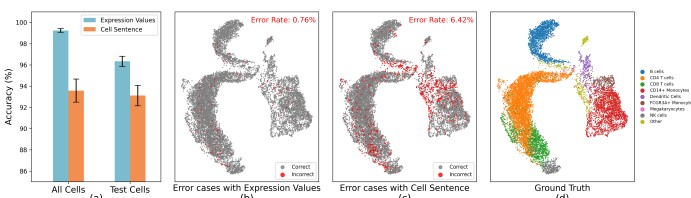

Figure 2: Comparison between a cell's scRNA-seq representation and its textual description.

Figure 3: (a) Representing cells as cell sentences leads to significant accuracy degradation. (b-d) UMAP vis. of predicted cell types and ground truth. Prediction with cell sentences yields lower scores.

and cell type–that is not directly captured by expression profiles but is critical for downstream biological interpretation. Incorporating such textual knowledge offers a promising path toward more comprehensive and semantically meaningful cell representations.

To harness the rich biological knowledge in text descriptions for single-cell analysis, recent studies have explored joint cell-text modeling (Zhao et al., 2024; Liu et al., 2023b; Bian et al., 2024; Fang et al., 2025). However, we find that these efforts overlook key aspects of language-enhanced cell representation learning: **insufficient unimodal cell representation learning**, and **incomplete cell-text alignment**.

To address these gaps, we propose Single-Cell Multi-modal Generative Pre-trained Transformer (**scMMGPT**), a language-enhanced cell representation learning framework for single-cell transcriptomics. Rather than exploring chat-based interfaces with text LLMs (Fang et al., 2025; Levine et al., 2024; Schaefer et al., 2024), scMMGPT emphasizes improved performance in essential single-cell analysis tasks with information from the text modality. To ensure **robust cell representation learning**, scMMGPT preserves critical quantitative gene expression information during tokenization and employs contrastive pre-training with physiologically-motivated data augmentations. To achieve **comprehensive cell-text alignment**, scMMGPT incorporates two separate projectors for the bidirectional information sharing between scLLM and text LLM, and a comprehensive two-stage pre-training strategy with both discriminative and generative objectives.

With these breakthroughs, scMMGPT significantly surpasses existing unimodal and multimodal methods, establishing a new standard for leveraging textual knowledge in single-cell transcriptomics analysis. As shown in Figure 1 and the experiment section, **we demonstrate scMMGPT's performance across nine datasets and five downstream tasks.** Notably, the model achieves relative improvements of **8.13%** for clustering on the COVID-19 dataset, **7.48%** for annotation on the Myeloid dataset, **2.93%** for perturbation prediction on the Adamson dataset, and **28.41%** for cell description generation on the immune tissue dataset. Further experiments demonstrate its improvement for zero-shot cell type annotation. Finally, we perform comprehensive ablation studies to validate the effectiveness of scMMGPT's key components.

## 2 RELATED WORKS

**Single-Cell LLMs.** Single-cell sequencing technologies provide diverse biological features that facilitate the interpretation of cellular structures and functions (Heumos et al., 2023; Cao & Gao, 2022). Early efforts in scRNA-seq analysis focused on statistical approaches such as Seurat (Satija et al., 2015) and Harmony (Korsunsky et al., 2019). Advances in scRNA-seq have also generated massive, high-precision transcriptomic datasets, driving the development of Single-Cell LLMs (scLLMs) (Ziegenhain et al., 2017). This technique quantifies the mRNA molecule abundance, producing gene expression matrices that record expression values of individual genes across cells (Ji et al., 2021). Previous works have developed transformer-based foundation models on scRNA-seq data, pre-training with masked learning objectives on millions of cells (Zhao et al., 2023; Theodoris et al., 2023; Yang et al., 2022; Hao et al., 2024). Subsequent works improve the learning process by incorporating cell labels, such as batch information (Cui et al., 2024) and other cell metadata (Bian et al., 2024). After fine-tuning, these LLMs have proven useful in practical downstream tasks such as cell-type annotation, cell clustering, and batch effect removal.

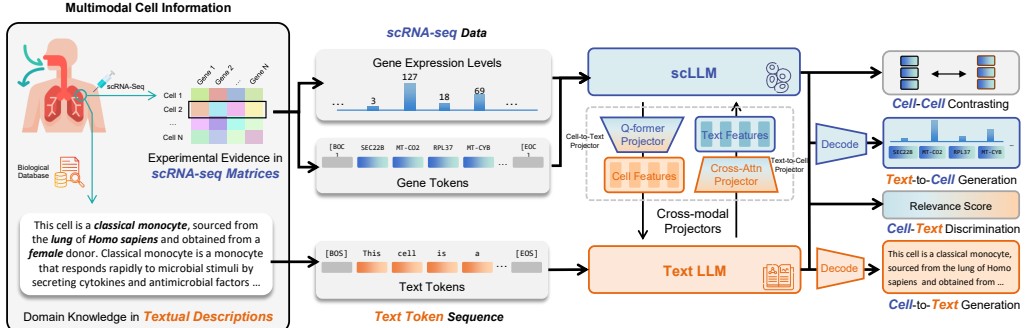

Figure 4: Overview of scMMGPT. (1) Cross-modal Discriminative Objective: Given paired cell and text inputs, the model learns to identify the correct textual description of a cell by aligning the outputs of the scLLM and text LLM. (2, 3) Cross-modal Generative Objectives: scMMGPT strengthens multimodal alignment through a unified generative pre-training strategy, jointly optimizing cell-to-text and text-to-cell translation tasks to facilitate bidirectional knowledge transfer.

**Cell-Text Modeling.** Incorporating free-text biological descriptions has proven useful for improving cellular representation learning, with prior works demonstrating that such auxiliary textual supervision can enhance the quality of cell embeddings (Chen & Zou, 2023; Liu et al., 2023b; Choi et al., 2024; Zhao et al., 2024). Beyond representation learning, enabling bidirectional translation between cells and text facilitates tasks such as universal cell-type annotation (Mao et al., 2025) and the generation of pseudo-cells (Luo et al., 2024a; Lopez et al., 2018). Recent efforts aim to build multimodal LLMs for single-cell data that align cell and text modalities directly (Fang et al., 2025; Schaefer et al., 2024). After instruction tuning on cell annotation or pseudo-cell generation, these models enable interactive single-cell analyses for human users. Nevertheless, the most crucial single-cell analysis tasks that boost real-world scientific discoveries rely more on efficient cell representation extraction to support tasks like cell annotation and clustering, which is often neglected in previous multimodal approaches.

**Scientific Multimodal LLMs.** Multimodal LLMs show remarkable potential for integrating data from various modalities (Li et al., 2023; Alayrac et al., 2022; Zhang et al., 2024), inspiring research for scientific modalities. Existing works have constructed multimodal LLMs for small molecules (Liu et al., 2023c; Fang et al., 2024; Liu et al., 2024a) and proteins (Xu et al., 2023b; Liu et al., 2024b) to tackle cross-modal scientific problems, such as description generation, molecular property prediction, and text-conditioned de novo design (Li et al., 2024; Edwards et al., 2022; Liu et al., 2023d; Cao et al., 2025; Liu et al., 2025; Luo et al., 2024b). Although single-cell analysis holds comparable scientific importance, the sparsity and high dimensionality nature of scRNA-seq data introduce a significant gap between transcriptomic and textual modalities, presenting challenges for joint modeling of them.

## 3 METHOD

To build scMMGPT, we first construct a diverse cell-text dataset to support cross-modal training (§3.1). Further, scMMGPT adopts a specialized scLLM that directly models original expression levels to bypass information loss, a pre-trained text LLM for generating descriptive annotations (§3.2), and bidirectional projectors to enable cross-modal information exchange. To facilitate comprehensive cell-text alignment and representation learning, we introduce a two-stage pre-training strategy (§3.3). After pre-training, scMMGPT can be applied to a range of downstream single-cell analysis tasks (§3.4). Figure 4 illustrates the overall architecture of scMMGPT.

### 3.1 LARGE SCALE AND MULTI-SOURCE CELL-TEXT DATA COLLECTION

**Large-Scale Single-Cell Transcriptomics Collection.** For single-cell transcriptomics data, we collect 60 million single-cell profiles from the biggest single-cell transcriptomics database CellxGene (Program et al., 2025). The collected cellular data includes high-resolution scNRA-seq matrices with gene names and numeric expression levels and associated cell properties such as cell types, tissues, and disease statuses. To ensure data quality, we then conduct data filtering and deduplication (see Appendix B for more details). After these steps, we maintain scRNA-seq data of 27 million cells

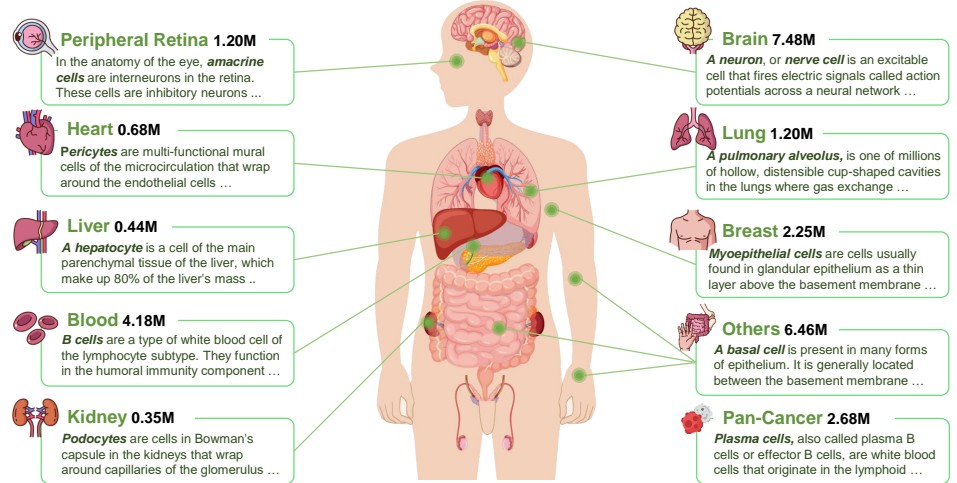

Figure 5: Summary of data used in the pre-training of scMMGPT. The dataset includes 27 million single-cell transcriptomic profiles from diverse human organs and tissues.

across various human tissues, summarized in Figure 5. The diverse cell atlas ensures generalization and prevents scMMGPT from degenerating into overfitting specific tissues.

**Large-Scale Cell Description Collection.** For textual information, we gather free-form cell identity explanation (*e.g.,* definition of cell types and diseases) from two sources: (1) the OBO Foundry (Smith et al., 2007), which integrates professional reference for biomedical terms, and (2) Wikipedia, which contains comprehensible explanations of cell function. We construct textual descriptions of cells by merging the free-form cell explanations and the cell metadata from CellxGenes.

## 3.2 NEURAL ARCHITECTURE FOR LANGUAGE-ENHANCED CELL REPRESENTATION LEARNING

To leverage the strengths of both modalities, we utilize pre-trained models for unimodal cell representation learning and natural language generation. These models bring in domain-specific knowledge and robust in-domain feature extraction capabilities.

**scLLM for Cell Representation Learning.** To obtain high-quality cell representations, we employ scGPT (Cui et al., 2024), a state-of-the-art single-cell language model pre-trained on large-scale scRNA-seq data. Its architecture is tailored to jointly model gene symbols and their quantitative expression levels, which avoids the loss of information that occurs with ranked gene lists.

Each cell is represented as a vector derived from the raw count matrix $\mathbf{X} \in \mathbb{N}^{N \times M}$, where $\mathbf{X}_{ij}$ denotes the expression level of gene $j$ in cell $i$. Since absolute gene expression values can vary across measurement platforms, we apply row-wise normalization before feeding into the model:

$$\widetilde{x}_j^{(i)} = \log\left(1 + \frac{x_j^{(i)}}{\sum_{k=1}^{M} x_k^{(i)}}\right). \tag{1}$$

To mitigate data sparsity, we retain the top 2,048 most expressed genes per cell, which captures the majority of meaningful signals.

**Text LLM for Text Generation.** For the text LLM, we utilize LLaMA-2 7B (Touvron et al., 2023b), a decoder-only transformer that excels at text generation. Its extensive pre-training and generative architecture make it well-suited for biomedical text understanding and generation, such as describing cellular states. We tokenize the textual description of cells into a sequence of tokens $\mathbf{t}^{(i)} = [t_1^{(i)}, t_2^{(i)}, \ldots, t_T^{(i)}]$ using the tokenizer in text LLM.

**Bi-directional Projectors for Cell-Text Translation.** To bridge the modality gap between cell and text LLMs, we introduce bidirectional cell-to-text and text-to-cell projectors that enable effective cross-modal alignment and information exchange.

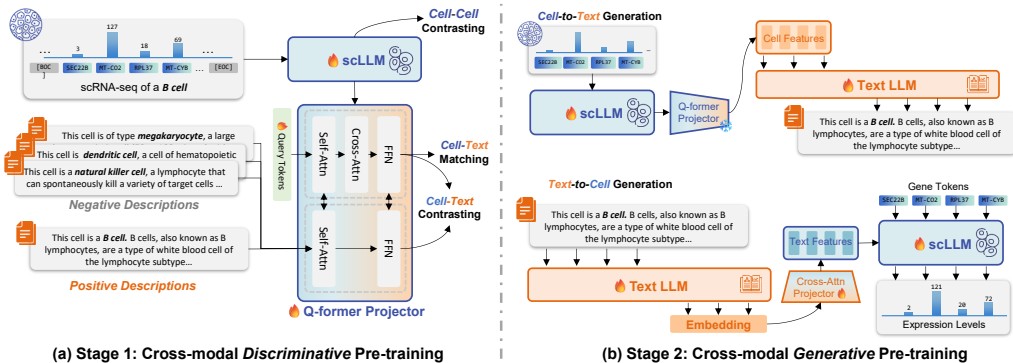

Figure 6: The two-stage cross-modal pre-training scheme. (a) In cross-modal discriminative pre-training, the model achieves cell-text integration by distinguishing matched cell-text pairs from unrelated pairs through contrastive and matching objectives. (b) In cross-modal generative pre-training, the model continues knowledge integration via unified generative tasks, including cell-to-text and text-to-cell generation objectives.

- **Cell-to-text projector** is implemented with a Query Transformer (Q-Former) (Li et al., 2023) with 32 learnable queries, which maps high-dimensional cell embeddings from the scLLM into the token embedding space of the Text LLM. We initialize the Q-Former with weights from BiomedBERT (Gu et al., 2021), a BERT encoder trained on PubMed abstracts and biomedical literature (Canese & Weis, 2013).
- **Text-to-cell projector** is realized using cross-attention layers (Vaswani et al., 2017) that project LLaMA-2 output embeddings into the hidden space of scGPT. The resulting representations are used as soft prompts (Li & Liang, 2021), conditioning the scLLM for text-to-cell generation.

## 3.3 Comprehensive Pretraining for Cell-Text Alignment and Cell Representation Learning

We adopt a two-stage pre-training strategy to pretrain cell representation, inject textual knowledge into the scLLM, align modalities, and enable bidirectional cell-text generation. Stage 1 aligns the representations and achieves coarse-grained knowledge injection through joint **cross-modal discriminative pre-training** and **contrastive cell representation learning.** Further, Stage 2 enables bi-directional cell-text translation via **generative pre-training**.

**Stage 1: Cross-modal discriminative pre-training.** We conduct cross-modal discriminative pre-training to create a shared latent space that captures semantic correspondences between scRNA-seq profiles and biomedical texts. Given a normalized single-cell expression vector $\widetilde{\mathbf{x}}^{(i)} \in \mathbb{R}^M$, the scLLM produces a contextualized embedding $\mathbf{h}_{\text{cell}} = \text{scGPT}(\widetilde{\mathbf{x}}^{(i)})$. This is passed through a Q-former to yield the cell feature $\mathbf{c} = \text{QFormer}(\mathbf{h}_{\text{cell}})$. Similarly, for a textual description represented by the token sequence $\mathbf{t}^{(\mathbf{i})} = \{t_1, \dots, t_L\}$, we extract the text embedding via the BERT module within the Q-Former as $\mathbf{h}_{text} = \text{BERT}(\mathbf{t}^{(i)})$.

During this stage, parameters of both the cell encoder and Q-Former are updated. We compute the alignment loss as a combination of the cell-text contrastive InfoNCE loss $\mathcal{L}_{\text{InfoNCE}}$ (Chen et al., 2020), and a cross-entropy cell-text matching loss $\mathcal{L}_{\text{CE}}$ (Li et al., 2023), thereby bringing paired representations closer while pushing unpaired ones apart:

$$\mathcal{L}_{\text{align}} = \mathcal{L}_{\text{InfoNCE}}(\mathbf{c}, \mathbf{t}) + \mathcal{L}_{\text{CE}}(\mathbf{c}, \mathbf{t}). \tag{2}$$

**Contrastive Cell Representation Learning.** In Stage 1, we perform joint contrastive learning along with the cross-modal discriminative pre-training for improved cell representation. Specifically, single-cell gene expression profiles are inherently noisy, suffering from high sparsity, dropout events, and technical variations caused by sequencing depth limits. Consequently, cells sharing the same biological state may exhibit significantly different raw expression values. To mitigate this issue and learn robust cell representations, we introduce a contrastive objective that leverages the intrinsic

invariance of scRNA-seq data. For cell profile $\mathbf{x}$, we generate two correlated views, $\mathbf{v}_1$ and $\mathbf{v}_2$, by applying a stochastic augmentation pipeline designed to simulate scRNA-seq technical artifacts:

- **Random Masking:** A random fraction of expressed genes are set to zero. This encourages the model to recover cell identity even when partial gene information is missing, mimicking the zero-inflation characteristic of scRNA-seq data.

- **Gaussian Jitter:** Multiplicative Gaussian noise is injected into non-zero gene values: $x_j \leftarrow x_j \cdot (1 + \epsilon)$, where $\epsilon \sim \mathcal{N}(0, \sigma^2)$. This ensures the representation is robust to small fluctuations in expression counts.

To align these views, we employ a contrastive objective that maximizes the agreement between latent representations of the same cell while distinguishing them from different cells in the batch. By treating augmented views of the same cell as positive pairs and all others as negatives, the objective forces the encoder to learn features invariant to stochastic perturbations. This process effectively filters out technical artifacts—such as dropout noise—and focuses the model on intrinsic biological signals. The resulting regularization yields a compact and robust latent manifold, providing a stable foundation for subsequent cross-modal integration.

**Stage 2: Cross-modal generative pre-training.** We further pre-train the model with cross-modal generative objectives to enhance the bidirectional knowledge injection:

- **Cell → Text Generation:** Given a cell embedding processed by the cell encoder and projected via the cell-to-text module, we condition the decoder-only text LLM to autoregressively generate corresponding textual descriptions. The objective is defined as:

$$\mathcal{L}_{\text{c2t}} = -\sum_{l=1}^{L} \log p(t_l \mid t_{<l}, \mathbf{c}),\tag{3}$$

  where $t_l$ denotes the $l$-th token of the generated text, and $\mathbf{c}$ represents the projected cell embedding.

- **Text → Cell Generation:** In the reverse direction, we enable the generation of cell embeddings conditioned on textual input. We first generate intermediate embedding with the text LLM: $\mathbf{c}' = \text{mlp}(\text{Llama}(t_{\leq L}))$. A lightweight text-to-cell projector then transforms this intermediate embedding into a soft prompt for the scLLM. The scLLM then predicts a pseudo-cell expression vector $\mathbf{x}' = \text{scGPT}(\mathbf{c}') \in \mathbb{R}^M$. We optimize the gene prediction head of scGPT with a mean squared error (MSE) loss:

$$\mathcal{L}_{\text{t2c}} = \sum_{1 \leq j \leq M} \text{MSE}(x'_j, \widetilde{x}_j),\tag{4}$$

  where $\widetilde{x}_j$ denotes the ground-truth normalized expression.

During this generative pre-training phase, we freeze the parameters of the cell-to-text projector and update the parameters of the text LLM, scLLM, and the text-to-cell projector by minimizing the combined loss $\mathcal{L}_{\text{c2t}} + \mathcal{L}_{\text{t2c}}$.

## 3.4 ADAPTING SCMMGPT TO DOWNSTREAM TASKS

After two-stage pre-training, scMMGPT can be applied to various single-cell analysis tasks, either in a zero-shot manner or with additional fine-tuning.

**Cell Type Annotation.** We leverage scMMGPT's ability to discern similarities between scRNA-seq profiles and cell-type text descriptions. In the zero-shot scenario, we directly use the stage-2 pre-trained scLLM and Q-former for annotation. Given a cell $\mathbf{x}^{(i)}$, we calculate its contrastive and matching losses against textual descriptions $\{\mathbf{t}_j\}$ of all possible cell types, and select the class that minimizes the combined loss $\lambda \mathcal{L}_{\text{InfoNCE}} + (1 - \lambda) \mathcal{L}_{\text{CE}}$ as prediction. In the fine-tuning scenario, we continue training scMMGPT on the downstream dataset using the alignment loss $\mathcal{L}_{\text{align}}$, and employ the same inference approach as in the zero-shot case.

**Batch Effect Correction and Cell Clustering.** These tasks evaluate the quality of cell representations derived from the model. Specifically, we leverage the fully pre-trained scLLM (*i.e.,* scGPT) from

Table 1: Results of cell type annotation (%) with fine-tuning. Asterisk (*) denotes results borrowed from previous studies (Cui et al., 2024; Liu et al., 2023b). **Bold** denotes best results. Green highlights relative improvements.

| Method | Myeloid | | hPancreas | | Multiple Sclerosis | | PBMC-3K | |
|---|---|---|---|---|---|---|---|---|
| | Accuracy (%) | F1 (%) | Accuracy (%) | F1 (%) | Accuracy (%) | F1 (%) | Accuracy (%) | F1 (%) |
| scBERT (Yang et al., 2022)* | 52.5 | 29.8 | 96.4 | 68.5 | 78.5 | 59.9 | 23.5 | 13.1 |
| scGPT (Cui et al., 2024)* | 64.2 | 34.6 | 96.8 | 71.8 | 85.6 | 70.3 | 93.3 | 80.7 |
| Geneformer (Theodoris et al., 2023) | 59.3 | 35.6 | 96.6 | 77.3 | 71.3 | 74.4 | 86.4 | 65.0 |
| LangCell (Zhao et al., 2024) | 58.9 | 35.7 | 96.3 | 70.8 | 72.9 | 71.2 | 90.6 | 81.2 |
| scELMo (Liu et al., 2023b)* | - | - | 96.8 | 68.0 | - | - | 90.3 | 83.5 |
| **scMMGPT** | **69.0**$^{+7.48\%}$ | **67.6**$^{+89.36\%}$ | **98.2**$^{+1.45\%}$ | **81.1**$^{+4.92\%}$ | **87.4**$^{+2.10\%}$ | **84.5**$^{+13.58\%}$ | **94.8**$^{+4.64\%}$ | **93.5**$^{+11.98\%}$ |

Table 2: Results of cell clustering on PBMC-10K (Gayoso et al., 2022) and COVID-19 (Lotfollahi et al., 2022b) datasets. Green highlights relative improvements.

| Dataset | Model | Avg$_{bio}$ (↑) | NMI$_{cell}$ (↑) | ARI$_{cell}$ (↑) | ASW$_{cell}$ (↑) | Avg$_{batch}$ (↑) | ASW$_{batch}$ (↑) | Graph$_{conn}$ (↑) |
|---|---|---|---|---|---|---|---|---|
| PBMC-10K | Seurat | 0.724 | 0.808 | 0.722 | 0.641 | 0.940 | 0.960 | 0.920 |
| | Harmony | 0.784 | 0.860 | 0.902 | 0.591 | 0.940 | 0.975 | 0.906 |
| | scVI | 0.753 | 0.819 | 0.847 | 0.592 | 0.947 | 0.967 | 0.928 |
| | scGPT | 0.821 | 0.850 | 0.873 | 0.740 | 0.923 | 0.950 | 0.895 |
| | Geneformer | 0.793 | 0.825 | 0.846 | 0.709 | - | 0.928 | - |
| | LangCell | 0.808 | 0.845 | 0.854 | 0.724 | - | 0.979 | - |
| | **scMMGPT** | **0.854**$^{+4.02\%}$ | **0.885**$^{+2.91\%}$ | **0.928**$^{+3.00\%}$ | **0.748**$^{+1.08\%}$ | **0.988**$^{+4.33\%}$ | **0.983**$^{+0.41\%}$ | **0.993**$^{+7.00\%}$ |
| COVID-19 | Seurat | 0.413 | 0.513 | 0.289 | 0.437 | 0.790 | 0.799 | 0.781 |
| | Harmony | 0.327 | 0.482 | 0.185 | 0.313 | 0.680 | 0.642 | 0.720 |
| | scVI | 0.502 | 0.638 | 0.408 | 0.461 | 0.838 | 0.833 | 0.844 |
| | scGPT | 0.504 | 0.659 | 0.400 | 0.452 | 0.850 | 0.826 | 0.874 |
| | **scMMGPT** | **0.545**$^{+8.13\%}$ | **0.668**$^{+1.37\%}$ | **0.454**$^{+11.27\%}$ | **0.512**$^{+11.06\%}$ | **0.892**$^{+4.94\%}$ | **0.875**$^{+5.04\%}$ | **0.908**$^{+3.89\%}$ |

both stages for zero-shot cell feature extraction. Formally, given a set of cells $\{\mathbf{x}^{(i)}\}$, we obtain their representations as $\mathbf{c}^{(i)} = \text{scGPT}(\mathbf{x}^{(i)})$, which are then utilized for clustering.

**Cell Description Generation.** For cell description generation tasks, we further fine-tune the text LLM with the cell-to-text translation loss $\mathcal{L}_{c2t}$, while freezing both the scLLM and the Q-former. During inference, we autoregressively generate descriptions conditioned on the cell embedding $\mathbf{c}$.

## 4 EXPERIMENTS

In experiment, we aim to answer these Research Questions (RQ):

**RQ1**: How effectively does scMMGPT enhance single-cell analysis tasks (cell type annotation, clustering, and text generation) by integrating biological priors from scRNA-seq and text?

**RQ2**: How do the discriminative and generative pre-training objectives synergistically contribute to the alignment and knowledge transfer between single-cell and text modalities?

**RQ3**: Does scMMGPT achieve robust generalization in challenging out-of-distribution scenarios?

**Experiment Setup.** scMMGPT is pre-trained in two stages: Stage 1 for representation alignment spans 5 epochs, while Stage 2 cross-modal generation runs for 1 epoch. Unless otherwise indicated, we apply LoRA (Hu et al., 2022) adapters for parameter-efficient fine-tuning of text LLM, whereas the cell encoder and projection modules are fully trained. Further details are provided in Appendix C.

### 4.1 EFFECTIVENESS FOR SINGLE-CELL ANALYSIS (RQ1)

Here we demonstrate the effectiveness of scMMGPT across four different tasks. The performance of the fifth tasks pseudocell generation can be found in Appendix D.2.

**Cell Type Annotation.** Cell type annotation is to evaluate a model's ability to accurately annotate cells based on their scRNA-seq profiles. We compare scMMGPT against several baselines: scBERT (Yang et al., 2022), scGPT (Cui et al., 2024), Geneformer (Theodoris et al., 2023), LangCell (Zhao et al., 2024), and scELMo (Liu et al., 2023b). Accuracy and macro F1 score (F1) are used as evaluation metrics. We conduct experiments on four datasets: Myeloid (Cheng et al., 2021), hPancreas (Chen et al., 2023), Multiple Sclerosis (Schirmer et al., 2019), and PBMC-3K (Zheng et al., 2017). All the models are fine-tuned on the downstream training dataset before the evaluation.

Table 3: Results of cell description generation on the immune tissue (Domínguez Conde et al., 2022) dataset.

| Model | Accuracy (↑) | F1 (↑) | BLEU-2 (↑) | ROUGE-2 (↑) | METEOR (↑) | MMD (↓) | EMD (↓) |
|---|---|---|---|---|---|---|---|
| GPT-2 Small (Radford et al., 2019) | 21.96% | 12.58% | 36.82% | 26.49% | 38.61% | 0.189 | 0.020 |
| GPT-2 Large (Radford et al., 2019) | 33.93% | 15.99% | 41.31% | 35.18% | 44.02% | 0.127 | 0.020 |
| LLaMA2-7B-Instruct (Touvron et al., 2023b) | 68.19% | 56.93% | 45.48% | 69.80% | 76.60% | 0.097 | 0.014 |
| C2S Small (Levine et al., 2024) | 35.05% | 25.67% | 50.07% | 47.53% | 55.38% | 0.043 | 0.016 |
| C2S Large (Levine et al., 2024) | 59.24% | 54.97% | 73.38% | 68.54% | 74.32% | 0.020 | 0.009 |
| **scMMGPT** | **87.56%**$^{+28.41\%}$ | **86.93%**$^{+52.70\%}$ | **91.45%**$^{+24.63\%}$ | **92.55%**$^{+32.59\%}$ | **90.72%**$^{+18.43\%}$ | **0.010**$^{-49.8\%}$ | **0.005**$^{-46.5\%}$ |

The experimental results are summarized in Table 1. Asterisk (*) denotes results borrowed from previous studies (Cui et al., 2024; Liu et al., 2023b). scMMGPT consistently outperforms existing methods across all evaluated datasets, achieving steady improvements in accuracy and up to about 10% increase in F1 scores (Multiple Sclerosis and PBMC-3K). These results validate scMMGPT's superior classification performance and its strong adaptability across different biological contexts and tissue distributions.

**Cell Clustering.** Cell clustering plays a fundamental role in novel cell type discovery and helps remove the batch effects in scRNA-seq data, which are introduced by different experiment batches. We evaluate scMMGPT on cell clustering tasks using the PBMC-10K (Gayoso et al., 2022) and COVID-19 (Lotfollahi et al., 2022b) datasets, comparing it against several scLLMs and softwares: Seurat (Satija et al., 2015), Harmony (Korsunsky et al., 2019), scVI (Lopez et al., 2018), scGPT (Cui et al., 2024), Geneformer (Theodoris et al., 2023), and LangCell (Zhao et al., 2024). We use standard biological conservation ($NMI_{cell}$, $ARI_{cell}$, and $ASW_{cell}$) and batch correction ($ASW_{batch}$ and $Graph_{Conn}$) metrics (Luecken et al., 2022) to assess performance, which can be summarized into $Avg_{bio}$ and $Avg_{batch}$ scores. The detailed explanations of these metrics are in Appendix C.3.

As shown in Table 2, scMMGPT consistently outperforms all baselines across both biological conservation and batch correction metrics. On both datasets, scMMGPT achieves an $Avg_{bio}$ improvement of $4\%$ and $8\%$, and an $Avg_{batch}$ improvement of at least $4\%$. These results highlight its strong ability to preserve biological structure while effectively correcting batch effects.

**Cell Description Generation.** Cell description generation is to produce cells' textual descriptions using scRNA-seq data. We compare against GPT-2 (Radford et al., 2019) and C2S (Levine et al., 2024) on the immune tissue (Domínguez Conde et al., 2022) dataset. We use Maximum Mean Discrepancy (MMD) and Earth Mover's Distance (EMD) to measure semantic similarity based on text embeddings (Xiao et al., 2024), alongside BLEU (Papineni et al., 2002), ROUGE (Lin, 2004), and METEOR (Banerjee & Lavie, 2005) to measure rule-based text similarity. We also measure the cell annotation accuracy and F1 scores based on the cell type text extracted from the cell descriptions.

As shown in Table 3, scMMGPT substantially outperforms prior baselines, achieving higher text similarity and semantic alignment with ground-truth descriptions. It improves classification metrics by nearly 30%, rule-based text similarities by about 20%, and reduces MMD and EMD by nearly 50%. These results indicate the strong biological relevance of the descriptions generated by scMMGPT.

**Perturbation Prediction.** We evaluate the perturbation prediction capability by predicting the post-perturbation gene expression profile given an input control cell state and the perturbed gene(s). Following the setting of scGPT (Cui et al., 2024), we fine-tune each model on a subset of perturbations and test it on perturbations involving unseen genes. We report $Pearson_\delta$, which measures the correlation between predicted and observed expression changes after per-

Table 4: Results of perturbation prediction on Adamson and Norman datasets. Higher is better.

| Dataset | Model | $Pearson_\delta$ (↑) | $Pearson_\delta$+de (↑) |
|---|---|---|---|
| Adamson | **scMMGPT** | **0.633**$^{+2.93\%}$ | **0.791**$^{+0.25\%}$ |
| | scGPT | 0.615 | 0.789 |
| | GEARS | 0.531 | 0.678 |
| | LR | 0.387 | 0.620 |
| Norman | **scMMGPT** | **0.587**$^{+0.69\%}$ | **0.752**$^{+1.35\%}$ |
| | scGPT | 0.583 | 0.742 |
| | GEARS | 0.547 | 0.715 |
| | LR | 0.387 | 0.620 |

turbation. We further report $Pearson_\delta$+de, computed on the top-20 differentially expressed genes for each perturbation. As Table 4 shows, scMMGPT consistently outperforms scGPT, GEARS (Roohani et al., 2024), and the linear regression (LR) baseline for both datasets.

Table 5: Ablation studies of cell type annotation on various datasets.

| Method | Myeloid | | hPancreas | | Multiple Sclerosis | | PBMC-3K | |
|---|---|---|---|---|---|---|---|---|
| | Accuracy | F1 | Accuracy | F1 | Accuracy | F1 | Accuracy | F1 |
| **scMMGPT** | **68.96** | **67.57** | **98.22** | **81.06** | 87.38 | **84.49** | **94.83** | **93.51** |
| *Ablation over Training Stages* | | | | | | | | |
| No Pre-train Stage 2 | 67.44 | 66.78 | 97.63 | 80.56 | **87.79** | 83.40 | 94.72 | 93.36 |
| No Pre-train Stage 1&2 | 67.56 | 63.22 | 77.13 | 44.03 | 84.98 | 76.45 | 94.72 | 91.41 |
| Stage 1 w/o $\mathcal{L}_{\text{InfoNCE}}$ | 64.21 | 64.43 | 93.62 | 68.94 | 86.41 | 81.22 | 94.50 | 93.17 |
| Stage 1 w/o $\mathcal{L}_{\text{CE}}$ | 68.26 | 63.06 | 96.70 | 74.52 | 86.82 | 82.77 | 94.61 | 92.04 |
| scLLM from scratch | 66.13 | 64.11 | 97.18 | 81.40 | 75.94 | 75.72 | 92.36 | 82.52 |
| *Ablation over generative objectives* | | | | | | | | |
| Stage2 w/o c2t | 67.42 | 67.15 | 96.92 | 79.96 | 86.55 | 83.69 | 94.29 | 92.86 |
| Stage2 w/o t2c | 67.74 | 66.74 | 97.11 | 79.93 | 87.32 | 82.75 | 94.50 | 93.14 |
| *Ablation over Cross-model Projector* | | | | | | | | |
| use MLP instead of Qformer | 66.60 | 64.51 | 96.73 | 74.41 | 86.23 | 83.38 | 94.72 | 91.92 |
| *Ablation over Text Source* | | | | | | | | |
| only cell metadata | 67.97 | 67.54 | 97.79 | 80.76 | 87.51 | 82.15 | 94.18 | 91.39 |
| only free text | 67.94 | 66.69 | 96.56 | 74.41 | 83.03 | 78.23 | 94.83 | 93.34 |

Table 6: Results of cell type annotation (%) on the Tabula Sapiens (Consortium* et al., 2022) dataset. The models are fine-tuned on a certain proportion of test cell types. Acc@N denotes top-N accuracy. Green highlights relative improvements.

| Model | Zero-Shot | | | Fine-tuned on 10% Types | | | Fine-tuned on 20% Types | | | Fine-tuned on 30% Types | | |
|---|---|---|---|---|---|---|---|---|---|---|---|---|
| | Acc@1 | Acc@5 | Acc@10 | Acc@1 | Acc@5 | Acc@10 | Acc@1 | Acc@5 | Acc@10 | Acc@1 | Acc@5 | Acc@10 |
| Random | 0.6 | 3.1 | 6.2 | 0.6 | 3.1 | 6.2 | 0.6 | 3.1 | 6.2 | 0.6 | 3.1 | 6.2 |
| BioTranlator | - | - | - | 3.5 | 33.6 | 45.4 | 13.4 | 48.2 | 63.5 | 13.7 | 50.6 | 68.6 |
| LangCell | 28.6 | 69.2 | 82.9 | 30.5 | 71.0 | 83.7 | 35.0 | 74.6 | 86.4 | 38.2 | 83.0 | 92.1 |
| **scMMGPT** | **49.1**$^{+71.7\%}$ | **83.1**$^{+19.9\%}$ | **91.1**$^{+9.9\%}$ | **55.7**$^{+82.6\%}$ | **89.2**$^{+25.6\%}$ | **96.0**$^{+14.7\%}$ | **59.7**$^{+71.1\%}$ | **90.4**$^{+21.2\%}$ | **96.8**$^{+12.0\%}$ | **60.9**$^{+59.4\%}$ | **93.6**$^{+12.8\%}$ | **98.4**$^{+7.0\%}$ |

## 4.2 ABLATION STUDIES (RQ2)

We conduct systematic ablation studies to evaluate the effectiveness of each key component.

**Impact of Training Stages.** We first assess the effectiveness of our multi-stage training process. We compare five different pre-training variations: (1) using the full pre-training pipeline, (2) using a randomly initialized scLLM instead of a pre-trained one, (3) removing $\mathcal{L}_{\text{InfoNCE}}$ or $\mathcal{L}_{\text{CE}}$ loss in stage 1 pre-training, (4) removing stage 2 pre-training, and (5) removing both stage 1&2 pre-training. As shown in Table 5, the full training pipeline yields the best results, with clear improvements in both accuracy and F1-score.

**Impact of Text Source.** We next investigate how different textual inputs affect performance. We evaluate three variations: (1) using both cell metadata and free-text descriptions as the text source, (2) using metadata only, and (3) using free-text only. As shown in Table 5, combining metadata and free-text leads to the highest accuracy and F1-score. This highlights that structured and unstructured textual information provides complementary benefits.

**Impact of Generative Objectives.** We further study the role of the two generative objectives in stage 2 pre-training. Specifically, we ablate (1) cell-to-text generation (Stage2 w/o c2t) and (2) text-to-cell generation (Stage2 w/o t2c). As Table 5 shows, both objectives contribute to cross-modal alignment.

**Impact of Model Architecture.** We examine the effect of the Q-former in capturing cross-modal interactions. We try an MLP variant of scMMGPT where the Q-former projector is replaced with simple MLP layers while keeping other components unchanged. As shown in Table 5, the original Q-former design outperforms the MLP variant across all metrics, highlighting its effectiveness in modeling fine-grained relationships between modalities.

## 4.3 OUT-OF-DISTRIBUTION EVALUATION (RQ3)

We evaluate the model's cell annotation performance under out-of-distribution settings. The experiment is conducted on the Tabula Sapiens (Consortium* et al., 2022) dataset, which comprises 161 distinct human cell types, most absent from our pre-training corpus. We report the accuracies

under fine-tuning settings, where the model is trained on different proportions of test cell types (10%, 20%, and 30%) and tested on the remaining proportions. We compare our model against two baseline methods, BioTranslator (Xu et al., 2023a) and LangCell (Zhao et al., 2024). Top-N accuracies (Acc@N) are used as the metric.

Experimental results are shown in Table 6. scMMGPT achieves an Acc@1 of 49.1% and an Acc@5 of 83.1% without further fine-tuning, surpassing certain fine-tuned baselines. As the proportion of fine-tuning cell types increases, scMMGPT consistently improves its performance across all metrics, reaching a maximum Acc@1 of 60.9% when fine-tuned on 30% cell types. These results demonstrate that scMMGPT's pre-trained knowledge of cellular and textual data enables strong generalization to unseen cell types.

## 5 CONCLUSION AND FUTURE WORKS

In this study, we introduce scMMGPT, a multimodal foundation model for comprehensive single-cell analysis. By effectively bridging scRNA-seq data with textual information, scMMGPT supports a range of tasks, including cell type annotation, cell clustering, perturbation prediction, cell description generation, and pseudo cell generation. This multi-modal integration is achieved through the synergistic use of an scLLM and a text PLM, connected by our innovative cross-modal pretraining. Trained on 27 million cells from the CELLxGENE dataset, scMMGPT demonstrates superior performance across diverse single-cell analysis applications.

Looking forward, we aim to expand scMMGPT by incorporating datasets of more species and integrating more cell modalities, including scATAC-seq and CITE-seq. This expansion will empower scMMGPT to address more challenges of multi-omic integration (Lotfollahi et al., 2022a), cross-omic translation (Liu et al., 2023a), and novel cell type discovery (Yang et al., 2022), significantly enhancing its utility in single-cell research.

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

## A    LIMITATIONS AND ETHICAL CONSIDERATIONS

**Limited coverage of non-human species.** A key limitation of scMMGPT is its reliance on pre-training data primarily sourced from the CELLxGENE dataset (Program et al., 2025), which focuses mainly on human tissues. This restricts the model's ability to generalize to cells from other species, such as those from widely used mouse datasets (Franzén et al., 2019).

**Lack of incorporation of multiomics.** Another major limitation is scMMGPT's exclusive focus on transcriptomic data, without incorporating other single-cell sequencing modalities such as scATAC-seq or CITE-seq (Liu et al., 2023a; Lin et al., 2022). By analyzing RNA abundance alone, the model misses critical insights into chromatin accessibility (scATAC-seq) and protein expression (CITE-seq). Integrating these modalities could provide a more comprehensive understanding of cellular states and regulatory mechanisms.

**Exclusion of spatial transcriptomics.** scMMGPT is restricted to analyzing individual cells, disregarding the spatial context. Spatially resolved transcriptomics captures gene expression and the physical organization of cells, thus revealing critical insights into cell-cell interactions and microenvironments (Wen et al., 2023b;a; Wang et al., 2025). Since scMMGPT operates solely on scRNA-seq data, it cannot leverage spatial relationships, which are often important in understanding tissue function and disease mechanisms.

**Ethical considerations.** Our pre-training and evaluation data are sourced from publicly available datasets, and we follow the original dataset licenses and usage terms; however, genomic data can still be sensitive, and users should ensure appropriate privacy protection, consent compliance, and access control when adapting the model to restricted-access or clinical data. The pre-training corpus is dominated by human tissues and a limited set of experimental protocols, which may introduce biases across populations, tissues, and sequencing platforms, and thus careful validation is recommended before deployment in underrepresented cohorts or new domains. Finally, while scMMGPT can accelerate biological analysis and drug discovery, it may be misused to support unsupported biomedical claims or downstream decisions without expert oversight; we emphasize that the model is intended for research use and should not be used as a standalone tool for clinical decision-making.

Table 7: Dataset statistics before and after data filtering.

| Tissue/Category | Pre-filtering | Post-filtering |
|---|---|---|
| Brain | 22 M | 7.5 M |
| Lung | 3.3 M | 1.2 M |
| Pancreas | 0.22 M | 0.08 M |
| Pan-cancer | 4.4 M | 2.6 M |
| Kidney | 1.0 M | 0.35 M |
| Heart | 2.2 M | 0.7 M |
| Blood | 5.4 M | 4.2 M |
| Others | 22 M | 10.3 M |
| Total | 60.5 M | 26.9 M |

## B    DETAILS OF DATASETS

### B.1    COLLECTION OF THE PRE-TRAINING DATASET

#### B.1.1    CELL TRANSCRIPTOMICS COLLECTION

The pre-training dataset for scMMGPT is constructed using publicly available data from the Cellx-Gene database (Program et al., 2025), with a snapshot taken on July 1, 2024. The dataset undergoes a series of filtering steps to ensure quality and consistency:

- We retain only human single-cell RNA sequencing (scRNA-seq) data, excluding entries from other species.

- We focus on data generated using the 10X Genomics platform, as its standardized outputs minimize technical variability across datasets.

- We deduplicate the dataset by keeping only one copy of each unique cell.

- To prevent information leakage, we remove all cells that appear in the test sets of downstream evaluation datasets.

After these filtering steps, the final dataset comprises approximately 27 million cells from 344 categories and 60697 different genes spanning diverse human tissues, including brain, lung, heart, blood, pancreas, kidney, pan-cancer, and others, as summarized in Figure 5. Table 7 shows the statistics of the dataset before and after the filtering.

### B.1.2 TEXTUAL DESCRIPTION COLLECTION

To ensure consistent and accurate cell-type annotations, we integrate standardized descriptions from two key resources: the Open Biomedical Ontologies Foundry (OBO Foundry) (Smith et al., 2007) and English Wikipedia. For each cell in the pre-training dataset, we first identify its biological classification (e.g., "Tendon Cell"). These classifications are then mapped to formal definitions in OBO Foundry's Cell Ontology, which provides machine-readable terms for cell types.

Additionally, we supplement these definitions with detailed explanations extracted from relevant Wikipedia entries, enriching the textual descriptions with accessible and comprehensive context.

---

**Example Cell Description from Wikipedia.**

**Tendon Cell:** Tendon cells, or tenocytes, are elongated fibroblast type cells. The cytoplasm is stretched between the collagen fibres of the tendon. They have a central cell nucleus with a prominent nucleolus. Tendon cells have a well-developed rough endoplasmic reticulum and they are responsible for synthesis and turnover of tendon fibres and ground substance.

Tendon cells form a connecting epithelial layer between the muscle and shell in molluscs. In gastropods, for example, the retractor muscles connect to the shell via tendon cells. Muscle cells are attached to the collagenous myo-tendon space via hemidesmosomes. The myo-tendon space is then attached to the base of the tendon cells via basal hemidesmosomes, while apical hemidesmosomes, which sit atop microvilli, attach the tendon cells to a thin layer of collagen. This is in turn attached to the shell via organic fibres which insert into the shell. Molluscan tendon cells appear columnar and contain a large basal cell nucleus. The cytoplasm is filled with granular endoplasmic reticulum and sparse golgi. Dense bundles of microfilaments run the length of the cell connecting the basal to the apical hemidesmosomes.

---

**Example Cell Description from the Open Biomedical Ontologies Foundry.**

**Tendon Cell:** An elongated fibrocyte that is part of a tendon. the cytoplasm is stretched between the collagen fibres of the tendon. they have a central cell nucleus with a prominent nucleolus. tendon cells have a well-developed rough endoplasmic reticulum and they are responsible for synthesis and turnover of tendon fibres and ground substance.

---

### B.2 COLLECTION OF DOWNSTREAM DATASET

We collected multiple benchmark datasets to evaluate the performance of the scMMGPT model in various downstream tasks.

- **CellxGene (Program et al., 2025)**: CellxGene is an interactive data portal for single-cell transcriptomic data. It provides a graphical user interface for exploring and analyzing standardized single-cell datasets. The platform supports functions such as dataset discovery, download, analysis, and annotation. It has been used in machine learning to get millions of cells.

- **PBMC-10K (Gayoso et al., 2022)**: Integrating two independent scRNA-seq studies of healthy human peripheral blood mononuclear cells, this resource captures 3,346 actively expressed genes across 9 defined cell types: B cells, CD4+/CD8+ T lymphocytes, CD14+/FCGR3A+ monocytes, dendritic cells, natural killer cells, megakaryocytes, and rare

populations. The dataset serves as a standardized benchmark for methodological validation in immunogenomics.

- **Human Pancreas (Chen et al., 2023)**: The human pancreas (hPancreas) dataset includes data from five scRNA-seq studies of human pancreas cells and is divided into two parts. The reference set comes from two data sources, and the query set includes the other three. The dataset covers 3,000 genes. The reference set contains 10,600 cells across 13 cell types: alpha, beta, ductal, acinar, delta, pancreatic stellate, pancreatic polypeptide, endothelial, macrophage, mast, epsilon, Schwann, and T cells. The query set contains 4,218 cells across 11 cell types.

- **Multiple Sclerosis (Schirmer et al., 2019)**: This dataset includes nine healthy control samples and twelve MS samples, following the scGPT (Cui et al., 2024) setup. We use the control samples as the reference set for model fine-tuning and keep the MS samples as the query set for evaluation. The reference set contains 7,844 cells, and the query set contains 13,468 cells. The original publication provided cell type labels, which we use as the ground truth for evaluation. The dataset contains 18 cell types and covers 3,000 highly expressed genes.

- **Myeloid (Cheng et al., 2021)**: The myeloid dataset includes nine different cancer types. Six of them are used in the reference set for training, and the remaining three are used in the query set. The reference set includes the cancer types UCEC, PAAD, THCA, LYM, cDC2, and kidney. The query set includes MYE, OV-FTC, and ESCA. The reference set has 9,748 cells across 21 cell types. The query set has 3,430 cells across 11 cell types. The dataset covers 3,000 genes with high expression values.

- **PBMC-3K (Zheng et al., 2017)**: This dataset contains 4,638 cell samples. It includes eight types of cells: B cells, CD4+/CD8+ T lymphocytes, CD14+/FCGR3A+ monocytes, dendritic cells, natural killer cells, and megakaryocytes. The PBMC-3K dataset is characterized by the analysis of 14,236 unique genes. The dataset is made up of two different batches that represent separate experimental conditions.

- **Immune Tissue (Domínguez Conde et al., 2022)**: This comprehensive reference dataset profiles 360,000 human immune cells through single-cell RNA sequencing (scRNA-seq), systematically annotated with 35 distinct cell subtypes. Derived from 16 tissue types across 12 adult donors, it provides a cross-tissue characterization of lymphocyte, myeloid, and stromal cell populations, establishing a baseline for immunological studies.

- **Tabula Sapiens (Consortium* et al., 2022)**: Spanning 24 human organs with 483,152 single-cell profiles, this pan-tissue atlas identifies 161 rigorously validated cell types across epithelial, immune, endothelial, and stromal lineages. Incorporating demographic diversity through multi-ethnic donors, it establishes transcriptional baselines from bladder mucosa to vascular endothelial cells using unified scRNA-seq protocols.

## C EXPERIMENTAL DETAILS

### C.1 PRE-TRAINING DETAILS

The scMMGPT model employs a multimodal pre-training framework that integrates gene expression data with textual information. Inheriting scGPT's (Cui et al., 2024) architecture, the cell encoder utilizes a gene vocabulary of 60,697 entries. For cellular input representation, we implement a top-value alignment strategy that selects the 2,048 highest-expressed genes along with their expression values. Cross-modal alignment is achieved through a Q-Former (Li et al., 2023) module with 32 query tokens, where the cross-attention mechanisms are activated every two layers.

Pre-training was executed on eight NVIDIA 4090D GPUs over five epochs (1.4 million total steps), requiring approximately five days for completion. The optimization process employed AdamW with a weight decay of 0.001 and a peak learning rate of $10^{-5}$, modulated through a linear warmup (1,000 steps from $10^{-6}$ minimum learning rate) followed by linear decay. We select 2 negative samples for each sample to calculate the InfoNCE (Oord et al., 2018) loss.

Table 8: Model architecture specifications

| Parameter | Value |
| --- | --- |
| Gene vocab size | 60,697 |
| Gene padding function | High value |
| Gene padding max len | 2,048 |
| QFormer BERT hidden dim | 768 |
| QFormer num_query_token | 32 |
| QFormer cross_attention_freq | 2 |
| Gene embed dim | 512 |
| Cell projector dim | 256 |
| Text projector dim | 256 |
| Language model hidden size | 2,048 |
| LM output max length | 128 |
| Cell decoder attention layer | 1 |
| Cell decoder attention head | 4 |

Table 9: Pre-training configurations

| Parameter | Value |
| --- | --- |
| Similarity function | Cosine similarity |
| Optimizer | AdamW |
| Scheduler | Linear |
| Max learning rate | 1e-05 |
| Warm up steps | 1000 |
| Weight decay | 0.001 |
| Batch size | 12 |

## C.2 DOWNSTREAM TRAINING DETAILS

For the fine-tuning of downstream tasks, we conduct single-epoch training with a constrained batch size of 4, preserving the AdamW optimizer configuration in the pre-training stage. Language model adaptation employs Low-Rank Adaptation (LoRA) (Hu et al., 2022) with a rank-decomposition dimension $r$ of 8, a scaling factor $\alpha$ of 32, and a dropout ratio of 0.1 for stochastic regularization during weight adaptation.

For each downstream analysis dataset, we perform quality control by removing the ambiguous categories (e.g., ''Other'', ''Unknown''). We establish symmetrical training pairs with strict 1:1 allocation between cellular generation and textual synthesis objectives. This balanced design promotes bidirectional cross-modal alignment while mitigating task dominance.

## C.3 METRIC DETAILS

For the evaluation of cell clustering, we use both biological conservation metrics and batch correction metrics (Luecken et al., 2022). All the metrics are the higher the better.

Biological conservation metrics:

- **NMI_cell** (Normalized Mutual Information): This metric measures the similarity between predicted clusters and ground-truth cell type labels.

- **ARI_cell** (Adjusted Rand Index): This metric measures the agreement between clustering results and true labels, adjusted for chance groupings.

- **ASW_cell** (Average Silhouette Width for cell types): This metric measures how well each cell fits within its assigned cluster compared to other clusters.

- **Avg_bio**: The average of the three biological conservation metrics NMI_cell, ARI_cell and ASW_cell.

Table 10: Ablation study on the ratio of positive/negative samples.

| Micro Batch Size | Negative Samples | Myeloid | | hPancreas | | Multiple Sclerosis | | PBMC-3K | |
|---|---|---|---|---|---|---|---|---|---|
| | | Accuracy | F1 | Accuracy | F1 | Accuracy | F1 | Accuracy | F1 |
| 3 (default) | 2 (default) | **69.45** | 65.62 | **94.97** | **71.19** | 87.12 | **83.25** | 94.72 | 91.84 |
| 6 | 5 | 68.25 | **65.74** | 91.28 | 68.27 | 86.45 | 82.73 | 94.61 | 91.07 |
| 12 | 11 | 66.68 | 65.04 | 86.84 | 62.61 | **87.6** | 82.7 | **95.04** | **92.55** |

Batch correction metrics:

- **$ASW_{batch}$** (Average Silhouette Width for batches): This metric measures the mixing of batches, where a lower silhouette score indicates better integration across batches.
- **$Graph_{conn}$** (Graph Connectivity): This metric measures how well cells from the same batch are connected in the nearest neighbor graph, indicating successful batch correction.
- **$Avg_{batch}$**: The average of the two batch correction metrics $ASW_{batch}$ and $Graph_{conn}$.

# D ADDICTIONAL EXPERIMENT RESULTS

## D.1 ABLATION STUDY ON THE RATIO OF POSITIVE/NEGATIVE SAMPLES

For the contrastive objective, the construction of positive and negative samples can affect the training dynamics. We vary the ratio between positive and negative samples by changing the micro-batch size and the number of negative samples. As shown in Table 10, different settings achieve the best performance on different benchmarks, while the overall differences remain moderate. This suggests that our method is robust to the positive/negative sampling ratio.

## D.2 TEXT-GUIDED PSEUDO-CELL GENERATION

We conduct cell generation experiments on the immune tissue (Domínguez Conde et al., 2022) dataset. We select several generative single-cell models as baselines, including scGen (Lotfollahi et al., 2019), scVI (Lopez et al., 2018), scDiffusion (Luo et al., 2024a), scGPT (Cui et al., 2024), and C2S (Levine et al., 2024). Inspired by previous studies, we train a simple $k$-Nearest Neighbors ($k$-NN) classifier on the test set to distinguish the generated cells. The classification accuracies under different $k$ values are reported to reflect the quality of the generated cells.

The results are presented in Table 11. scMMGPT achieves state-of-the-art performance in text-conditioned pseudo-cell generation, significantly outperforming all baseline models across all $k$-NN accuracies ($k$=3,5,10,25). The consistently high accuracy and low standard deviations of scMMGPT demonstrate its robustness and effectiveness in bridging cellular and textual data.

Table 11: Results of text-conditioned pseudo-cell generation on the immune tissue dataset. The baseline results are borrowed from (Levine et al., 2024).

| Model | $k$-NN Accuracy | | | |
|---|---|---|---|---|
| | $k = 3$ | $k = 5$ | $k = 10$ | $k = 25$ |
| scGEN (Lotfollahi et al., 2019) | 0.2376 ± 0.0112 | 0.2330 ± 0.0093 | 0.2377 ± 0.0053 | 0.2335 ± 0.0041 |
| scVI (Lopez et al., 2018) | 0.2436 ± 0.0062 | 0.2400 ± 0.0064 | 0.2425 ± 0.0034 | 0.2348 ± 0.0032 |
| scDiffusion (Luo et al., 2024a) | 0.2335 ± 0.0125 | 0.2288 ± 0.0111 | 0.2368 ± 0.0067 | 0.2306 ± 0.0049 |
| scGPT (Cui et al., 2024) | 0.1838 ± 0.0086 | 0.1788 ± 0.0169 | 0.1811 ± 0.0149 | 0.1882 ± 0.0071 |
| C2S (Levine et al., 2024) | 0.2588 ± 0.0061 | 0.2565 ± 0.0060 | 0.2746 ± 0.0073 | 0.2715 ± 0.0070 |
| **scMMGPT** | **0.2996 ± 0.0065**[+0.04] | **0.2992 ± 0.0055**[+0.04] | **0.2986 ± 0.0038**[+0.02] | **0.2981 ± 0.0051**[+0.03] |

## D.3 ROBUSTNESS IN CELL TYPE ANNOTATION

During the inference of cell type annotation, we combine the cell-text constructive InfoNCE loss and cell-text matching CE loss to get the type probability: $\lambda \mathcal{L}_{InfoNCE} + (1 - \lambda)\mathcal{L}_{CE}$ (§3.4). In this case,

the choice of $\lambda$ may influence the downstream annotation performance. We conduct inference on different levels of $lambda$ to test scMMGPT's robustness, and the results are shown in Table 12.

Table 12: The impact of lambda setting on our model.

| $\lambda$ | Myeloid | | hPancreas | | Multiple Sclerosis | | PBMC-3K | |
|---|---|---|---|---|---|---|---|---|
| | Accuracy | F1 | Accuracy | F1 | Accuracy | F1 | Accuracy | F1 |
| 0 | **68.96** | **67.57** | 97.58 | 80.52 | 87.28 | 83.8 | 94.29 | 92.92 |
| 0.01 | 68.79 | 67.49 | 97.56 | 80.48 | 87.29 | 83.87 | 94.29 | 92.92 |
| 0.05 | 67.71 | 65.5 | 97.56 | 81.06 | 87.4 | 83.93 | 94.4 | 92.98 |
| 0.1 | 67.01 | 64.15 | **97.63** | **81.49** | 87.35 | 83.91 | 94.4 | 92.98 |
| 0.2 | 65.81 | 62.76 | 97.51 | 80.54 | 87.35 | 83.85 | 94.4 | 93.07 |
| 0.3 | 64.96 | 61.55 | 97.41 | 80.45 | 87.33 | 84.38 | 94.72 | 93.45 |
| 0.4 | 64.5 | 60.77 | 97.44 | 80.46 | 87.38 | 84.49 | 94.72 | 93.45 |
| 0.5 | 64.29 | 60.24 | 97.41 | 80.42 | 87.42 | **84.5** | 94.72 | 93.45 |
| 0.6 | 64.12 | 59.93 | 97.41 | 80.52 | **87.43** | 83.95 | 94.72 | 93.45 |
| 0.7 | 63.77 | 59.54 | 97.41 | 80.72 | 87.39 | 83.83 | **94.83** | **93.51** |
| 0.8 | 63.45 | 59.24 | 97.41 | 80.72 | 87.42 | 83.53 | 94.83 | 93.51 |
| 0.9 | 63.39 | 59.2 | 97.39 | 80.68 | 87.38 | 83.24 | 94.83 | 93.51 |
| 0.95 | 63.33 | 59.12 | 97.39 | 80.68 | 87.38 | 81.67 | 94.83 | 93.51 |
| 0.99 | 63.24 | 59.05 | 97.39 | 80.68 | 87.38 | 81.67 | 94.83 | 93.51 |
| 1 | 63.24 | 59.05 | 97.39 | 80.68 | 87.37 | 81.66 | 94.83 | 93.51 |

## E  VISUALIZATION

### E.1  EFFECTIVENESS OF DIFFERENT CELL REPRESENTATION METHODS

To further quantify the information loss in cell sentences, we conduct a visualization experiment comparing cell sentence inputs with original expression values. Specifically, we train two separate MLPs with identical hyperparameters for cell type annotation on the PBMC-10K (Gayoso et al., 2022) dataset. As shown in Figure 3, the cell sentence representation leads to a significant increase in error rate, particularly when distinguishing morphologically similar cell types such as dendritic cells and FCGR3A+ monocytes. This finding highlights the non-negligible cellular information lost during the transformation from numerical expression levels to cell sentences, which limits the effectiveness of related models in downstream applications.

### E.2  BATCH EFFECT MITIGATION IN SCMMGPT EMBEDDINGS.

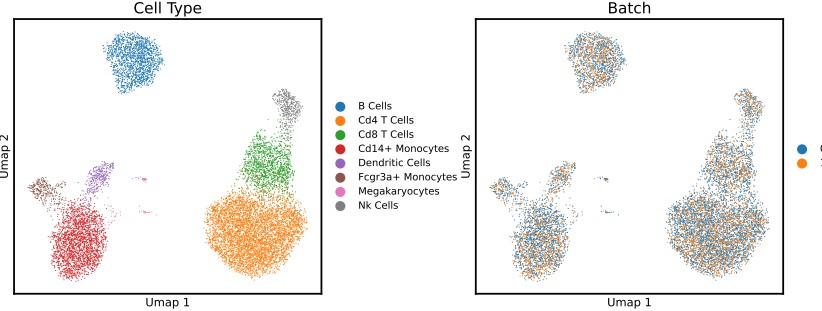

Figure 7: UMAP visualization of scMMGPT's embeddings for cells from different experimental batches on PBMC-10K (Gayoso et al., 2022). The result demonstrates the model's ability to capture cell type distinctions while effectively mitigating batch effects.

In wet lab experiments, it is challenging to maintain identical experimental conditions across different batches, which can lead to variations in the measured scRNA-seq data. We compute scMMGPT embeddings on PBMC-10K and visualize them using UMAP, as shown in Figure 7. The results

demonstrate that cell embeddings from scMMGPT effectively capture cell type differences while minimizing the influence of batch effects.

### E.3 BIOLOGICAL SIGNALS IN SCMMGPT EMBEDDINGS

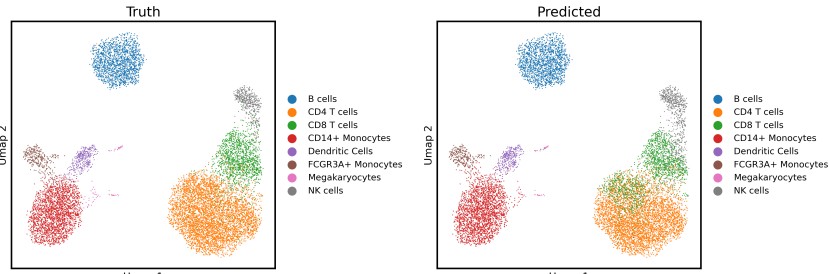

Figure 8: UMAP plot of embeddings from scMMGPT for PBMC-10K (Gayoso et al., 2022) dataset in zero-shot task. Plots are colored by actual cell type labels and predicted cell type labels.

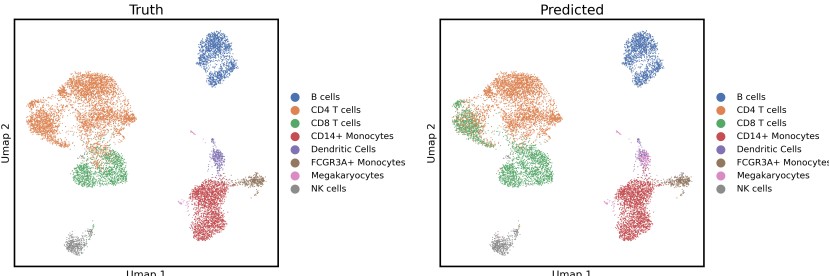

Figure 9: UMAP plot of embeddings from LangCell (Zhao et al., 2024) for PBMC-10K (Gayoso et al., 2022) dataset in zero-shot task. Plots are colored by actual cell type labels and predicted cell type labels.

We perform visualization on the PBMC10K dataset to show the cell embedding quality of scMMGPT (Figure 8). We also visualize the embeddings from the LangCell model (Figure 9) for comparison. In the plots, cells of the same type cluster closely together. When we compare the truth labels and predicted labels, we find that scMMGPT can correctly annotate most cells without fine-tuning.

### E.4 DETAILED DESCRIPTIONS FOR METRICS IN FIGURE 1.

In this section, we provide detailed descriptions and sources for each metric presented in Figure 1. Models not evaluated on a specific metric are represented by 0.

- **Myeloid, hPancreas, Multiple Sclerosis, PBMC-3K:** Data sourced from Table 1. We report the F1 score of each model for the cell type annotation task on these datasets.
- **PBMC-10K, Covid-19:** Data sourced from Table 2. We report the $\text{Avg}_{\text{bio}}$ of each model for the Cell cluster task.
- **Immune Tissue:** Data sourced from Table 11. We report the F1 score of each model for the cell description generation task on the Immune Tissue dataset.
- **Tabula Sapiens:** Data sourced from Table 6. We report the Acc@1 of each model on the Tabula Sapiens dataset under the "Fine-tuned on 30% Types" condition.
- **IT Gen:** Data sourced from Table 11. We report the 3-NN Accuracy of each model for the text-conditioned pseudo-cell generation task on the Immune Tissue dataset.

### E.5 HEATMAP OF CELL TYPE ANNOTATION RESULT

In this section, we show heatmaps of confusion matrices across different cell-type annotation datasets. We normalize each cell type to provide a clear and consistent view of the accuracy of the model. As

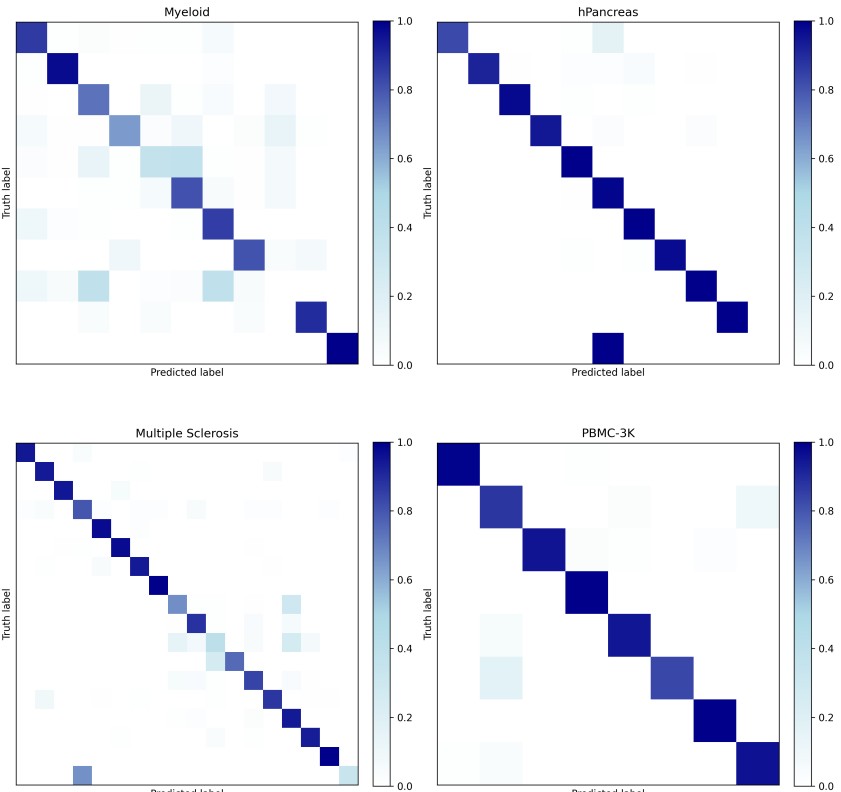

Figure 10: Heatmap of the results from scMMGPT's Cell Type Annotation on Myeloid (Cheng et al., 2021), hPancreas (Chen et al., 2023), Multiple Sclerosis (Schirmer et al., 2019) and PBMC-3k (Zheng et al., 2017) dataset.

shown in Figure 10, a clear diagonal line is visible in each heatmap, showing that our model achieves high prediction accuracy in all datasets.

### E.6    VISUALIZATION OF SCRNA-SEQ DATA.

To facilitate a better understanding of scRNA-seq matrices, we select a subset of cells from the Tabula Sapiens dataset for visualization. In wet-lab single-cell sequencing experiments, researchers measure the expression levels of a predefined set of genes across individual cells. Each value in the matrix represents the expression level of a corresponding gene within a single cell. The colors in the heatmap indicate the log1p-transformed expression levels.

### E.7    CELL-TEXT SIMILARITY MATRIX FOR ALL CELL TYPES

We visualize the cell-text similarity matrix across all cell types from multiple tissues, including blood, heart, brain, kidney, lung, pan-cancer, and pancreas. As shown in Figure 12, cell-text pairs from the same tissue exhibit higher similarity, indicating that our representations capture tissue-specific biological semantics. Moreover, the diagonal entries achieve the highest similarity scores, suggesting that each cell type is most similar to its corresponding text description.

## F    LICENSES FOR EXISTING ASSETS

In this section, we discuss the licenses and terms of use of the open-sourced assets involved in the development of scMMGPT.

- The CellxGene database is protected under CC-BY 4.0 license.
- The code and checkpoints of scGPT are under the MIT license.

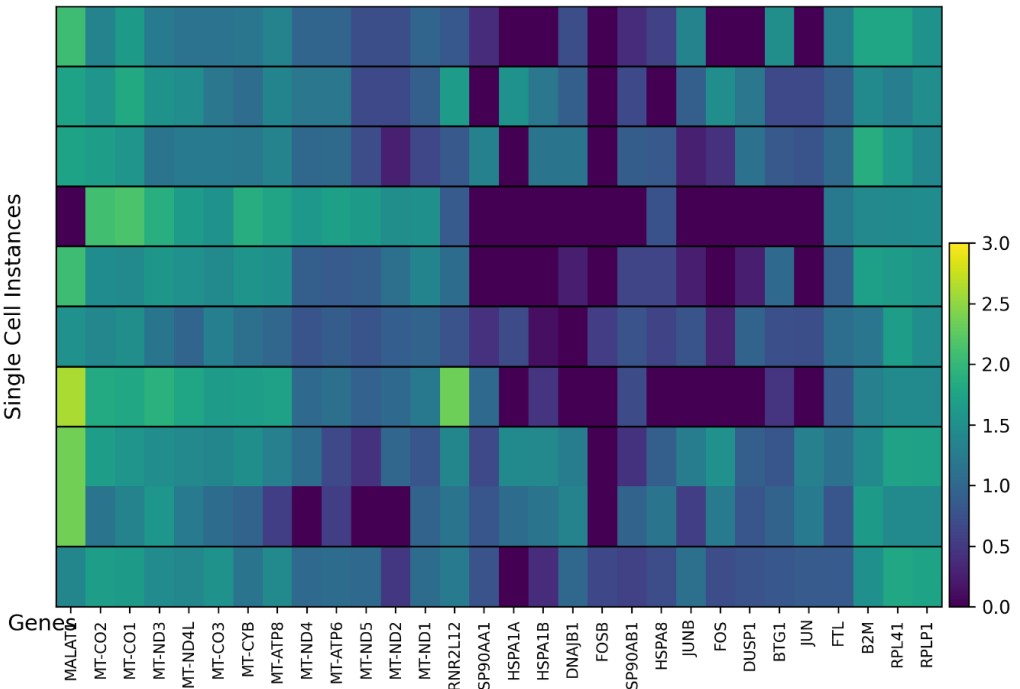

Figure 11: Visualization of a single-cell RNA sequencing matrix. Rows represent individual cells, and columns represent genes. The color intensity corresponds to the log1p-transformed expression levels, with darker shades indicating higher expression.

- The checkpoints of BioMedBert (Gu et al., 2021) are under the MIT license.
- The llama 2 series models are under the llama2 license.

The assets of this work are under the CC BY-NC license.

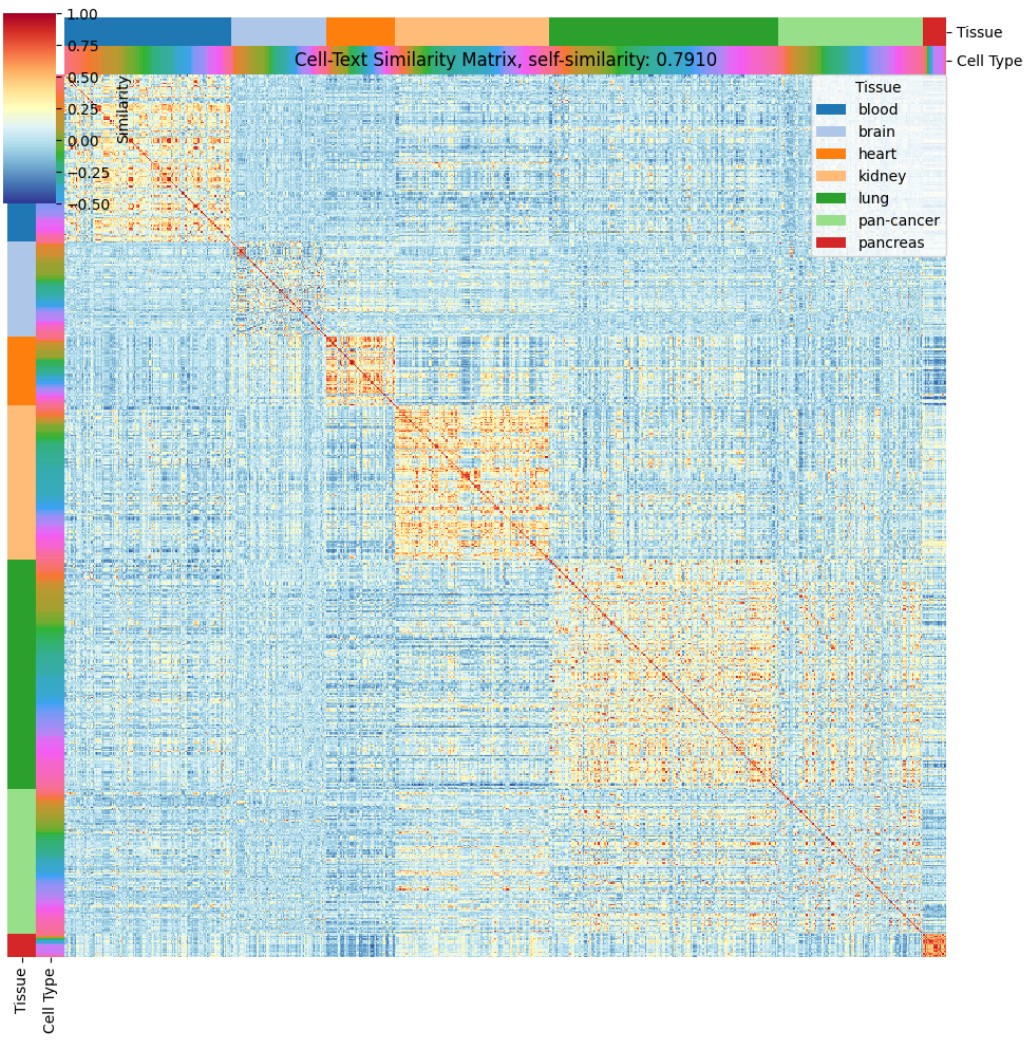

Figure 12: Cell-text similarity matrix for all cell types across blood, heart, brain, kidney, lung, pan-cancer, and pancreas tissues. Higher similarity is observed within the same tissue, and the diagonal exhibits the highest cell-text similarity.

