# OpenReview forum: "Language-Enhanced Representation Learning for Single-Cell Transcriptomics"
_ICLR.cc/2026/Workshop/FM4Science — ICLR 2026 Workshop FM4Science Poster_

### Official Review · Reviewer_poB5 · 2026-02-19
**scMMGPT model shows clear improvement over existing models and suggests a novel cross-modal pre training scheme, justified with extensive ablation study.  Strong accept**

**Rating:** 9
**Confidence:** 4

**Review:**

The paper is well organized and clearly states the motivation for the language-enhanced multimodality model. The strong results are convincing and consistent, and the ablation study is well structured. The two-step pre-training that uses joint embedding followed by generative steps is particularly interesting and might also be theoretically justified. As this is a non-trivial choice, I would recommend elaborating on the intuition behind using it. That would complement the experimental results and ablation study.
Additional comments
 - The text-to-cell channel was shown to improve general performance, but was not used explicitly in any downstream task. Did the authors test the generated cell sequences? Also, the MSE objective is probably naive and results in averaged predictions. Did the authors investigate more sophisticated generative heads/objectives?
 - The authors don't mention the uncertainty of the predictions. This is particularly important for the reliability of text-to-cell generation but also for any other scientific output. Do the authors have ideas on how to assess uncertainty at different stages of the model?
 - Contrastive Cell Representation Learning - the authors state that they used a contrastive objective. Is this simCLR loss? An explicit expression would be useful here.

---

### Official Review · Reviewer_3ZpV · 2026-02-19
**Review for Language-Enhanced Representation Learning for Single-Cell Transcriptomics**

**Rating:** 7
**Confidence:** 4

**Review:**

This paper introduces scMMGPT, a multimodal framework combining scRNA-seq models with text LLMs for single-cell representation learning. The method integrates an scRNA-seq encoder, a text LLM (LLaMA-2), and bidirectional projectors with a two-stage training pipeline involving contrastive alignment and cross-modal generation. The authors evaluate the model across multiple single-cell tasks and report consistent gains over unimodal and multimodal baselines.  This is a strong empirical multimodal bio-LLM paper with substantial evaluation and engineering effort. Overall clarity is good but dense. The methodological novelty is moderate.


Pros
- The paper well addressed that integrating text knowledge with omics data is important.
- The architecture of the model is reasonably sophisticated. The two-stage training (contrastive alignment + generative translation) is conceptually clean and easy to follow.
- Training leverages tens of millions of cells and textual metadata, which strengthens the foundation model framing.
- This work includes comprehensive empirical evaluation, e.g. annotation, clustering, perturbation prediction, generation.
- This work provides a strong ablation study.
- The results clearly show the consistent improvements across annotation and clustering tasks.

Cons
- While the system is large and complex, the conceptual novelty is limited. Multimodal alignment via projectors is well-established. Similar design patterns exist in multimodal LLM literature.
- There is a heavy reliance on existing components e.g. LLaMa2 text model, Q-former projectors, scGPT backbones etc. The novelty is mainly in integration rather than methodology.
- Several baselines rely on reported numbers from prior studies, making it unclear whether all models are evaluated under fully identical preprocessing and fine-tuning conditions
- Foundation model claim is debatable. Though the model is multimodal but still heavily fine-tuned for specific tasks.

---

### Official Review · Reviewer_wPLW · 2026-02-21
**Promising improvement of multimodal foundation learning of sc-seq integrating language and scRNA-seq representations**

**Rating:** 8
**Confidence:** 5

**Review:**

The authors continue the research using GPT to interpret and extract biological insights from sc-seq data using GPT, which also achieved SOTA results across benchmarks. In the technical solidness, the authors exhibited strong engineering integration with clear methodological explanation and visualization. Although GPT has previously applied on sc-seq analytics, a couple of improvements/novelty were achieved in this paper. The main contributions include: 1.) a systematic multimodal foundation model for sc-seq analysis by integrating scRNA-seq data and biological text into a single pre-trained framework 2.) multimodal-alignment for quantitative gene expression using contrastive learning. To sum up, there are a couple of things I suggest the authors to further polish the paper. This paper shed lights for future research directions.
Major concerns:
1. I am still unclear what the motivation or real-world application of text-to-cell generation?
2. Given the fact that knowledge comes from cell ontology and meta, how to make sure the model truly learn the biological semantics?
3. Tradeoff computational cost and efficiency is needed (FLOPs? Scaling analytics?). From my perspective, 27M parameters are huge.
Minor concerns
1.  54–63 The textual example of a monocyte is helpful. I may suggest integrate to figure 4.
2. L69 “insufficient unimodal cell representation learning.” This is vague.
3. The limitations section is appreciated. The authors may also mention potential bias introduced by curated ontology definitions influencing embedding alignment.
4. Line 182. Why threhold is 2048? What is the reasoning for setting up this number?
5. The ablation removing Stage 2 shows relatively small drops on some datasets. A short interpretation is needed.

---

### Meta-Review · Area_Chair_xfpt · 2026-02-28

**Recommendation:** Accept (Oral)
**Confidence:** 4

**Metareview:**

This paper presents scMMGPT, a multimodal framework integrating scRNA-seq representations with biological text through contrastive alignment and cross-modal generative training. Reviewers agree the work is technically sound, well engineered, and extensively evaluated across multiple datasets and tasks. The two-stage training strategy and systematic ablation study are particularly strong. While the conceptual novelty lies primarily in integration rather than new modeling primitives, the empirical gains are consistent and the contribution is highly relevant to foundation models for biology.

---

### Decision · Program_Chairs · 2026-03-03

Accept (Poster)